# Promoted cobalt metal catalysts suitable for the production of lower olefins from natural gas

Jingxiu Xie [1], Pasi P. Paalanen[1], Tom W. van Deelen [1], Bert M. Weckhuysen [1], Manuel J. Louwerse[1] & Krijn P. de Jong[1]

Due to the surge of natural gas production, feedstocks for chemicals shift towards lighter hydrocarbons, particularly methane. The success of a Gas-to-Chemicals process via synthesis gas (CO and $H_2$) depends on the ability of catalysts to suppress methane and carbon dioxide formation. We designed a Co/Mn/Na/S catalyst, which gives rise to negligible Water-Gas-Shift activity and a hydrocarbon product spectrum deviating from the Anderson–Schulz–Flory distribution. At 240 °C and 1 bar, it shows a $C_2$-$C_4$ olefins selectivity of 54%. At 10 bar, it displays 30% and 59% selectivities towards lower olefins and fuels, respectively. The spent catalyst consists of 10 nm Co nanoparticles with hcp Co metal phase. We propose a synergistic effect of Na plus S, which act as electronic promoters on the Co surface, thus improving selectivities towards lower olefins and fuels while largely reducing methane and carbon dioxide formation.

[1] Inorganic Chemistry and Catalysis, Debye Institute for Nanomaterial Science, Utrecht University, Universiteitsweg 99, 3584 CG Utrecht, The Netherlands. Correspondence and requests for materials should be addressed to K.P.d.J. (email: K.P.deJong@uu.nl)

The abundant availability of methane feedstock due to the shale gas revolution decreases the dependence on crude oil, however new technologies have to be developed to utilize its potential[1,2]. Methane may be converted to synthesis gas (syngas, a mixture of $H_2$ and CO), which can then be used to produce chemicals and fuels via the Fischer-Tropsch synthesis (FTS) process[3]. FTS is a surface polymerization reaction so the product selectivity is governed by the Anderson–Schulz–Flory (ASF) distribution[4]. Deviation of the ASF distribution to suppress methane formation is critical to attain high fractions of lower olefins (ethylene, propylene, and butylenes), and this is possible with promoted Fe-carbide-based[5–7] and promoted Co-carbide-based catalysts[8,9]. However, most carbide-based catalysts are also active for the water-gas-shift (WGS) reaction[10], thereby producing $CO_2$ and rendering them inefficient for methane-derived $H_2$-rich syngas. Similarly, the bifunctional oxide-zeolite catalysts, which convert syngas directly to lower olefins, showed high activity for WGS and are thus only suitable for CO-rich syngas[11,12]. The importance of decreasing $CO_2$ production during the FT step was recently highlighted by Wang et al. in their development of phase pure, stable and low-$CO_2$ selective ε-iron carbide FT catalysts for the coal-to-liquids process[13].

To be active for FTS but not for WGS, Co has to be in the metallic state during catalysis. Metallic Co-based catalysts are used commercially for the gas-to-liquids process in which long-chain saturated hydrocarbon products are produced that are subsequently cracked to valuable transportation fuels in particular kerosene and diesel[14–17]. The direct production of lower olefins from $H_2$-rich syngas is advocated, but this poses two challenges, specifically the suppression of methane and of $CO_2$ formation during FTS.

Till now, Co-based catalysts for the direct conversion of syngas to lower olefins focused on MnO as promoter, but the product spectrum was still dictated by the ASF distribution[18–22]. Adding alkali promoters to Co/MnO catalysts stimulates formation of Co-carbide, which inhibits methane, but promotes $CO_2$ production[23]. Besides acting as structural promoters, alkali metals were established to decrease activity for metallic Co-based catalysts and it was proposed to be correlated to the element electronegativity[24,25]. These alkali metals including Na or K, exist as oxides $Na_2O$ or $K_2O$ during catalysis, yet the oxygen counter-ion was often overlooked. The importance of counter-ions to alkali metal promoters was demonstrated previously for Fe-based catalysts[26], particularly the combination of Na and S was found to give a synergistic effect[27–29]. S is generally perceived to be a poison for Co-based catalysts in terms of activity and selectivity towards long-chain hydrocarbons ($C_{5+}$)[30], however it was also shown to decrease chain growth probability and improve olefins selectivity depending on its concentration[31–33]. Nonetheless, the influence of alkali metal and its counter-ion has not been considered for Co-based catalysts.

In this work, we demonstrate that the presence of Na plus S inhibits WGS and suppresses methane formation for metallic Co-based catalysts. We present an efficient metallic Co-based catalyst consisting of Co/Mn/Na/S, which has a product spectrum deviating from ASF distribution yet is inactive for WGS. The catalytic performance of this catalyst is evaluated over a range of reaction temperatures, 240–280 °C, and reaction pressures, 1–10 bar. $H_2$/CO feed ratio is kept constant at 2, a stoichiometric ratio relevant for methane feedstock. At industrially relevant conditions of 240 °C and 10 bar, $Co_1Mn_3$–$Na_2S$ shows superior product selectivities towards lower olefins and fuels in comparison to other Co-based catalysts. Detailed characterization of the spent catalysts using X-ray diffraction (XRD) and transmission electron microscopy (TEM) reveal 10 nm Co nanoparticles with hcp Co metal phase. Preliminary DFT calculations indicate the importance of the counter-ion for sodium and the consequences to catalysis. The approach of dispersing metallic Co nanoparticles on the MnO support, and utilizing alkali metal Na and its counter-ion S as electronic promoters is effective in reducing $CO_2$ and methane formation, hence creating new opportunities in gas-to-chemicals processes.

## Results

**Catalysts.** $Co_1Mn_3$ catalysts with an atomic ratio Co/Mn = 1/3 were synthesized via co-precipitation, and the calcined catalysts were impregnated with $Na_2CO_3$, $(NH_4)_2SO_4$, $Na_2S_2O_3$ or $Na_2S$ precursors followed by another calcination step. These catalysts were named $Co_1Mn_3$–$Na_2O$, $Co_1Mn_3$,–$SO_4^{2-}$, $Co_1Mn_3$–$Na_2S_2O_3$ and $Co_1Mn_3$–$Na_2S$, respectively. As a comparison, $Co_3Mn_1$ catalysts were also synthesized and named in a similar fashion. An overview of calcined catalysts and their elemental loadings of Mn, Co, Na and S are included in Supplementary Table 1. The XRD pattern of calcined $Co_1Mn_3$–$Na_2S$ (Supplementary Figure 1) consisted of $Mn_2O_3$, $MnO_2$ and $CoMnO_3$ phases, and the addition of promoters did not result in change of crystalline phases. An SEM image (Supplementary Figure 2) of calcined $Co_1Mn_3$–$Na_2S$, showed its morphology and the homogeneity of Co and Mn elemental loadings was confirmed by scanning electron microscopy-energy-dispersive X-ray spectroscopy (SEM-EDX, Supplementary Table 2). Scanning transmission electron microscopy-energy-dispersive X-ray spectroscopy (STEM-EDX) mapping (Supplementary Figure 3) also showed mixing of Co and Mn, and no isolated Co nanoparticles were observed.

**Catalytic performance.** Catalytic performance was evaluated at a range of reaction conditions (240–280 °C, 1–10 bar, $H_2$/CO = 2). At mild conditions of 240 °C, 1 bar, $H_2$/CO = 2, 1% CO conversion, $Co_1Mn_3$–$Na_2S$ displayed a high $C_2$–$C_4$ olefins selectivity of 54% with a $C_2$–$C_4$ olefin/paraffin ratio of 17. Moreover, methane selectivity at 17% was lower than what was predicted by the ASF distribution (Supplementary Figure 4 and Supplementary Table 3). While the addition of $Na_2S$ improved selectivity, it also decreased activity which is in agreement with literature that S is detrimental to activity of metallic Co catalysts[30,32]. In a control experiment, addition of sulfur only (without Na) was shown to decrease activity, while increasing methane selectivity (Supplementary Figure 4 and Supplementary Table 3).

The effects of reaction pressures and temperatures on the catalytic performance of $Co_1Mn_3$–$Na_2S$ are shown in Fig. 1 and detailed information can be found in Supplementary Tables 4–7. From Fig. 1a, at 10 bar, $H_2$/CO = 2, 13–30% CO conversion, an increase in temperature from 240 to 280 °C corresponded to decrease in $C_2$–$C_4$ olefins and $C_{5+}$ selectivities, but increase in methane and $C_2$–$C_4$ paraffins selectivities. For $Co_1Mn_3$–$Na_2S$ 21% $CO_2$ selectivity was attained at 280 °C and 10 bar. From Fig. 1b, at 240 °C, $H_2$/CO = 2, 10–18% CO conversion, an increase in pressure from 3 to 10 bar corresponded to an increase in $C_{5+}$ selectivity together with a decrease of selectivity towards $C_1$–$C_4$ hydrocarbon products. For $Co_1Mn_3$–$Na_2S$, no $CO_2$ production was detected at 3–10 bar, 240 °C, $H_2$/CO = 2. The increase in chain growth probability, α, due to increase in pressure was confirmed by the ASF distribution plot in Supplementary Figure 5. Notably, the methane fraction was always lower than expected from the ASF distribution for $Co_1Mn_3$–$Na_2S$. Since a high olefin/paraffin ratio was attained and no $C_1$ olefin exists, a lowered $C_1$ fraction is to be expected. Nonetheless, only with $Na_2S$ promotion this is actually achieved, while in literature catalysts always produce more methane than expected.

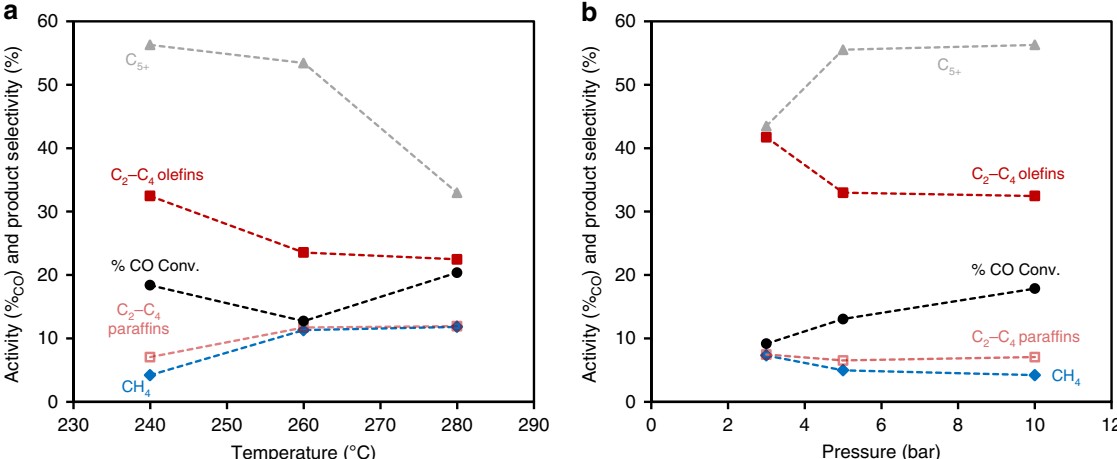

**Fig. 1** Catalytic performance of $Co_1Mn_3$–$Na_2S$ at different reaction temperatures or pressures. **a** Activity and selectivity at 240–280 °C, 10 bar, and $H_2/CO$ = 2, and **b** activity and selectivity at 240 °C, 3–10 bar, and $H_2/CO$ = 2. Activity is shown here as % CO conversion and product selectivity is shown in terms of methane, $CH_4$ (blue solid diamonds), $C_2$–$C_4$ olefins (red solid squares), $C_2$–$C_4$ paraffins (light red open squares), and $C_{5+}$ (grey solid triangles) which include all other products except $CO_2$ and $C_1$–$C_4$ hydrocarbons

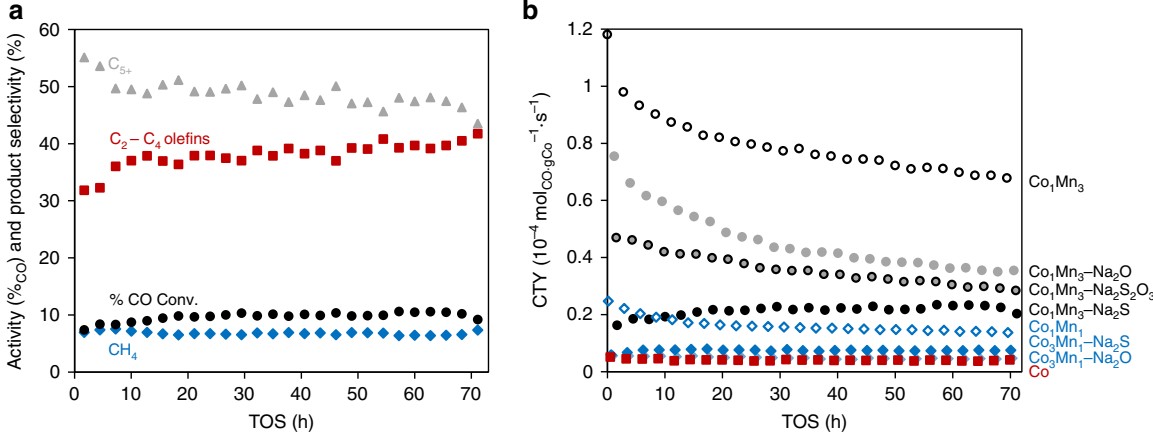

**Fig. 2** Catalytic performance over 70 h time-on-stream. Reaction conditions: 240 °C, 3 bar, and $H_2/CO$ = 2. **a** Activity in terms of %CO conversion (black solid circles) and selectivity towards methane, $CH_4$ (blue solids diamonds), $C_2$–$C_4$ olefins (red solid squares), and $C_{5+}$ (grey solid triangles) of $Co_1Mn_3$–$Na_2S$ over time, and **b** activity in terms of cobalt-time-yield, CTY, of various Co-based catalysts, namely $Co_1Mn_3$ (black open circles), $Co_1Mn_3$–$Na_2O$ (grey solid circles), $Co_1Mn_3$–$Na_2S_2O_3$ (grey solid with black outline circles), $Co_1Mn_3$–$Na_2S$ (black solids circles), $Co_3Mn_1$ (blue open diamonds), $Co_3Mn_1$–$Na_2S$ (blue solid diamonds), $Co_3Mn_1$–$Na_2O$ (light blue solid diamonds), and bulk Co (red squares) over time

Catalytic stability is an important consideration hence the catalytic performance of $Co_1Mn_3$–$Na_2S$ over 70 h is shown in Fig. 2a. The activity of $Co_1Mn_3$–$Na_2S$ showed an initial increase and remained then constant over 70 h. Methane selectivity remained stable over time, while $C_{5+}$ (all products except $CO_2$ and $C_1$–$C_4$ hydrocarbons) and $C_2$–$C_4$ olefin selectivities also stabilized after 10 h. The activity and stability of $Co_1Mn_3$–$Na_2S$ were also compared with other Co-based catalysts in Fig. 2b. As shown in Fig. 2b, the addition of Mn increased activity for Co-based catalysts, which is in agreement with literature[34,35]. Catalysts with Co/Mn ≈0.3 showed highest activity per gram Co (cobalt-time-yield, CTY), and the addition of $Na_2O$, $Na_2S_2O_3$ or $Na_2S$ decreased activity. Nonetheless, the activity of $Co_1Mn_3$–$Na_2S$ was still higher than the remaining Co-based catalysts. In terms of stability, $Co_1Mn_3$, $Co_1Mn_3$–$Na_2O$, $Co_1Mn_3$–$Na_2S_2O_3$ and $Co_3Mn_1$ showed deactivation but all other catalysts remained stable over 70 h.

At more industrially relevant conditions of 240 °C, 10 bar, $H_2/CO$ = 2, 18–30% CO conversion, the catalytic performance of $Co_1Mn_3$–$Na_2S$ was compared to other Co-based catalysts

(Table 1). $Co_1Mn_3$–$Na_2S$ displayed the highest selectivity towards lower olefins and lowest selectivities towards undesired methane and lower paraffins ($C_2$–$C_4$ olefin/paraffin ratio = 4.2). Remarkably, $CO_2$ selectivity was below 3% (below detection limit, see Supplementary Figure 6 for chromatograms), suggesting the absence of WGS activity and making it an attractive catalyst for $H_2$-rich syngas. $CO_2$ selectivity was consistently below detection limit for all catalysts except where less Mn is present, i.e. $Co_3Mn_1$–$Na_2O$ and $Co_3Mn_1$–$Na_2S$. Even so, the suppression of WGS activity by $Na_2S$ instead of $Na_2O$ addition was evident by the $CO_2$ selectivity of $Co_3Mn_1$–$Na_2S$ compared to $Co_3Mn_1$–$Na_2O$, i.e. 13 versus 28%, respectively. The precursor of Na/S and loading of Na were varied ($Na_2S$ and $Na_2S_2O_3$) and the favourable effects on selectivity remain (Supplementary Table 1 and Table 1). Further optimization of precursor and loadings of the promoters is however outside the scope of this study.

Bulk Co catalyst had the highest α, therefore its main product was $C_{5+}$ hydrocarbons. The addition of Mn–Co resulted in a lower α, and the addition of $Na_2O$ or $Na_2S$ further decreased α. Bulk Co catalyst showed typical ASF distribution deviation,

**Table 1 Catalytic performance at 240 °C, 10 bar, $H_2/CO = 2$, 18–30% CO conversion**

| | CO conv., $X$ (%) | CTY ($10^{-4}$ $mol_{CO} \cdot g_{Co}^{-1} s^{-1}$) | $C_1$, $S$ (%) | $C_2$–$C_4$ olefins, $S$ (%) | $C_2$–$C_4$ paraffins, $S$ (%) | $C_{5+}$, $S$ (%) | $CO_2$, $S$ (%) | O/P $C_2$–$C_4$ | $\alpha$ |
|---|---|---|---|---|---|---|---|---|---|
| Co | 32 | 0.13 | 12 | 6 | 7 | 75 | <2 | 0.8 | 0.69 |
| $Co_3Mn_1$ | 31 | 0.14 | 11 | 17 | 10 | 62 | <2 | 1.8 | 0.63 |
| $Co_3Mn_1$–$Na_2O$ | 20 | 0.09 | 9 | 14 | 7 | 42 | 28 | 1.9 | 0.53 |
| $Co_3Mn_1$–$Na_2S$ | 25 | 0.12 | 5 | 20 | 7 | 56 | 13 | 2.9 | 0.56 |
| $Co_1Mn_3$ | 31 | 0.56 | 15 | 12 | 13 | 61 | <2 | 0.9 | 0.67 |
| $Co_1Mn_3$–$Na_2O$ | 27 | 0.46 | 14 | 11 | 12 | 63 | <2 | 0.9 | 0.65 |
| $Co_1Mn_3$–$Na_2S$ | 18 | 0.40 | 4 | 30 | 7 | 59 | <3 | 4.2 | 0.53 |
| $Co_1Mn_3$–$Na_2S_2O_3$ | 22 | 0.42 | 7 | 25 | 12 | 56 | <3 | 2.1 | 0.50 |

CO conversion ($X$, %), activity per gram of Co (CTY), product selectivity ($S$, %), The detection limit of $CO_2$ selectivity is 0.5% yield, equivalent to 3% $CO_2$ selectivity at 18% CO conversion.

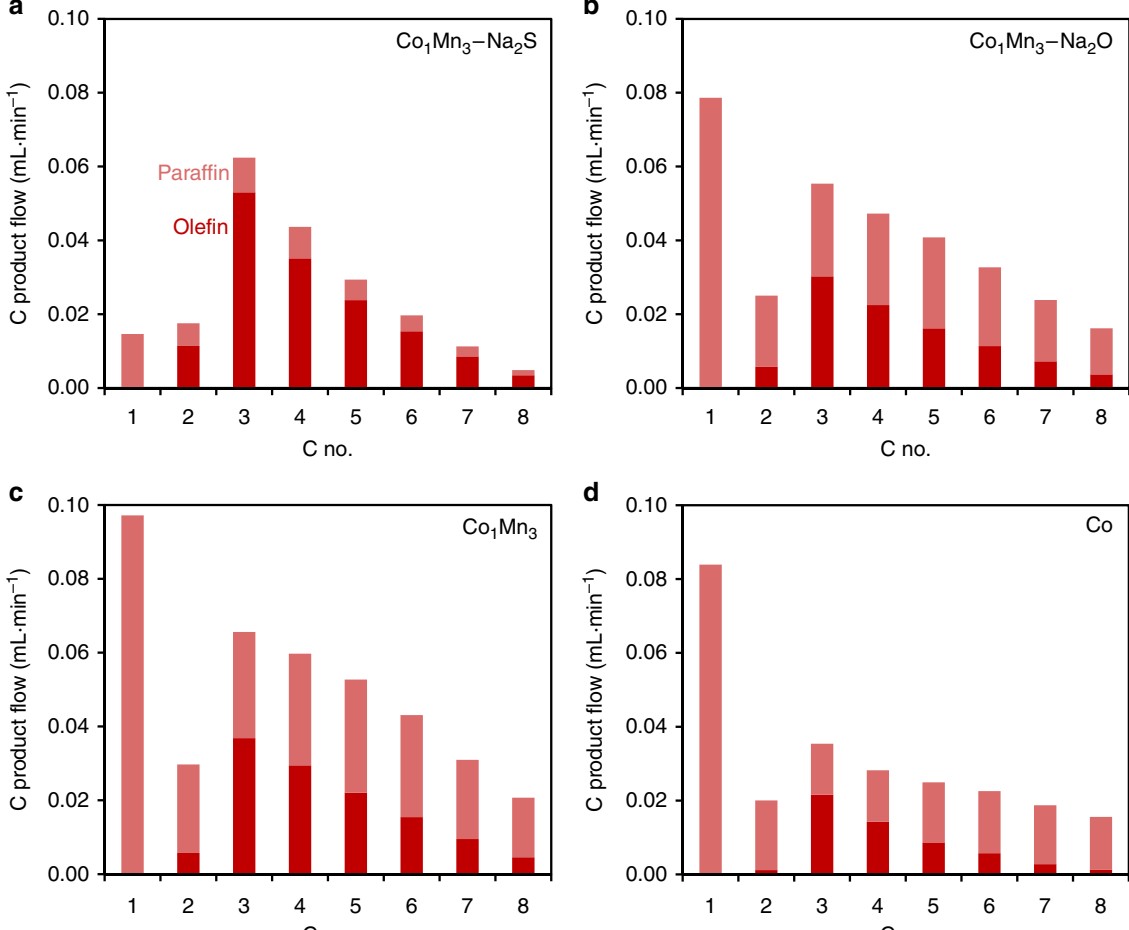

**Fig. 3** Distribution of 1-olefins and $n$-paraffins of $C_1$–$C_8$ hydrocarbon products. Reaction conditions: 240 °C, 10 bar, $H_2/CO = 2$, 18–30% CO conversion. **a** $Co_1Mn_3$–$Na_2S$, **b** $Co_1Mn_3$–$Na_2O$, **c** $Co_1Mn_3$ and **d** Co. Red bar corresponds to olefin product flow and light red bar corresponds to paraffin product flow

whereby the $C_1$ fraction is higher and the $C_2$ fraction is lower than predicted. The addition of $Na_2O$ suppressed the $C_1$ fraction, but this suppression was most prominent with the addition of $Na_2S$ (Supplementary Figure 5).

To obtain further mechanistic insights into the various catalytic systems, the detailed C product flow of 1-olefin and $n$-paraffin for each C number product is shown in Fig. 3. The mechanistic considerations of metallic Co FT catalysts include chain growth, chain branching, primary olefin/paraffin formation and olefin secondary reactions, such as secondary hydrogenation and isomerization[36]. From Fig. 3, $Co_1Mn_3$–$Na_2S$ produced significantly more primary olefins than linear paraffins for each C

containing hydrocarbon product. This suggests that β-H elimination was the dominant termination pathway for $Co_1Mn_3$–$Na_2S$ and secondary hydrogenation of olefins was also suppressed. Besides, the lower fraction of 2-butene in the $C_4$ hydrocarbon product spectrum of $Co_1Mn_3$–$Na_2S$ implied the suppression of secondary isomerization of olefins (Supplementary Table 9 and Supplementary Figure 7). This is in agreement with the presumption that secondary hydrogenation and isomerization of olefins take place at identical sites[36]. In addition, the low methane and $C_2$ hydrocarbon products from $Co_1Mn_3$–$Na_2S$ point to the blocking of sites for surface methyl, methylene and H species[37].

**Structure analysis of spent catalysts**. In order to understand the catalytic performance, the spent catalysts after being exposed to industrially relevant FTS conditions were characterized. Fig. 4a compares the XRD patterns of spent $Co_1Mn_3$, $Co_1Mn_3–Na_2S$ and $Co_3Mn_1–Na_2O$ and their crystalline phase compositions are summarized in Fig. 4b. Additional Rietveld QPA results for the spent catalysts are given in Supplementary Table 10. The diffraction patterns of crystallized wax were observed in Fig. 4a, and the wax present on the spent catalysts served to prevent oxidation of the spent catalysts. $Co_1Mn_3$ and $Co_1Mn_3–Na_2S$ consisted predominantly of a $Mn_{0.95}O$ phase, and a mixed $Mn_xCo_yO_4$ phase was observed which both contributed most likely not to any form of FT activity. Crucially, the hexagonal (hcp) metallic Co phase was present in both spent $Co_1Mn_3$ and $Co_1Mn_3–Na_2S$. The

average crystallite size for the hcp Co phase was 9.2 nm with a standard deviation of 1.9 nm. Small contributions from a $MnCO_3$ phase were also noted in both spent $Co_1Mn_3$ and $Co_3Mn_1–Na_2O$. In addition to the $Mn_{0.95}O$, $Mn_xCo_yO_4$, Co (hcp), $MnCO_3$ phases, a $Co_2C$ phase was present in spent $Co_3Mn_1–Na_2O$ in line with the work of Sun et al[8].

Fig. 5 shows the electron microscopy images and particle size distribution of spent $Co_1Mn_3–Na_2S$ after industrially relevant conditions (240–280 °C, 10 bar, and $H_2/CO = 2$), and STEM-EDX mappings were carried out to differentiate Co and Mn. From Fig. 5a, wax/amorphous carbon (indicated with arrows) was observed, which is in agreement with the XRD analysis in Fig. 4a. The Co particle size distribution from TEM revealed the average Co particle size to be 9.6 nm with a standard deviation of 4.4 nm,

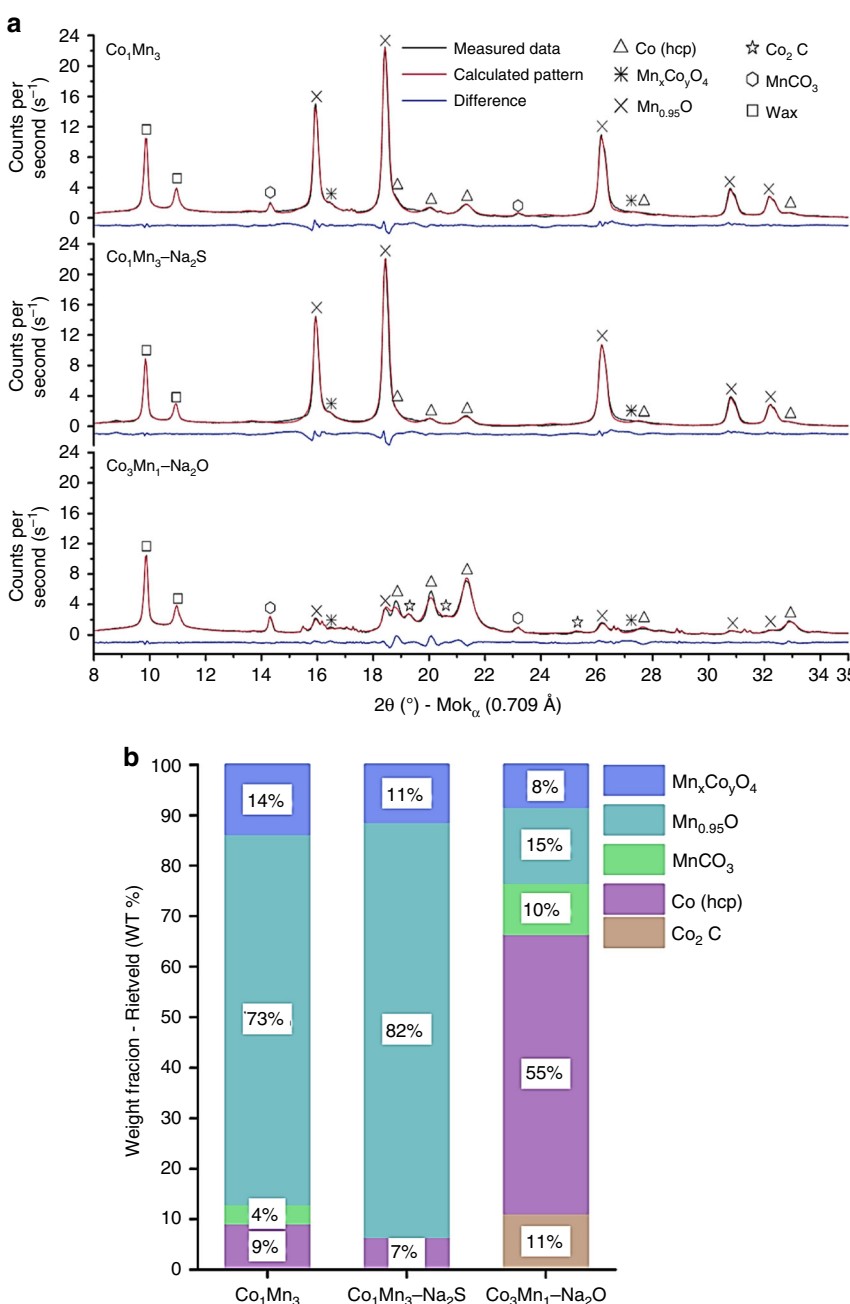

**Fig. 4** XRD analysis of spent $Co_1Mn_3$, $Co_1Mn_3–Na_2S$, $Co_3Mn_1–Na_2O$. Reaction conditions: 240–280 °C, 10 bar, and $H_2/CO = 2$. **a** Background corrected XRD patterns and **b** rietveld QPA-based crystalline phase compositions, which shows the $Mn_xCo_yO_4$ phase (blue), $Mn_{0.95}O$ (cyan), $MnCO_3$ (green), hcp Co (violet) and $Co_2C$ (brown)

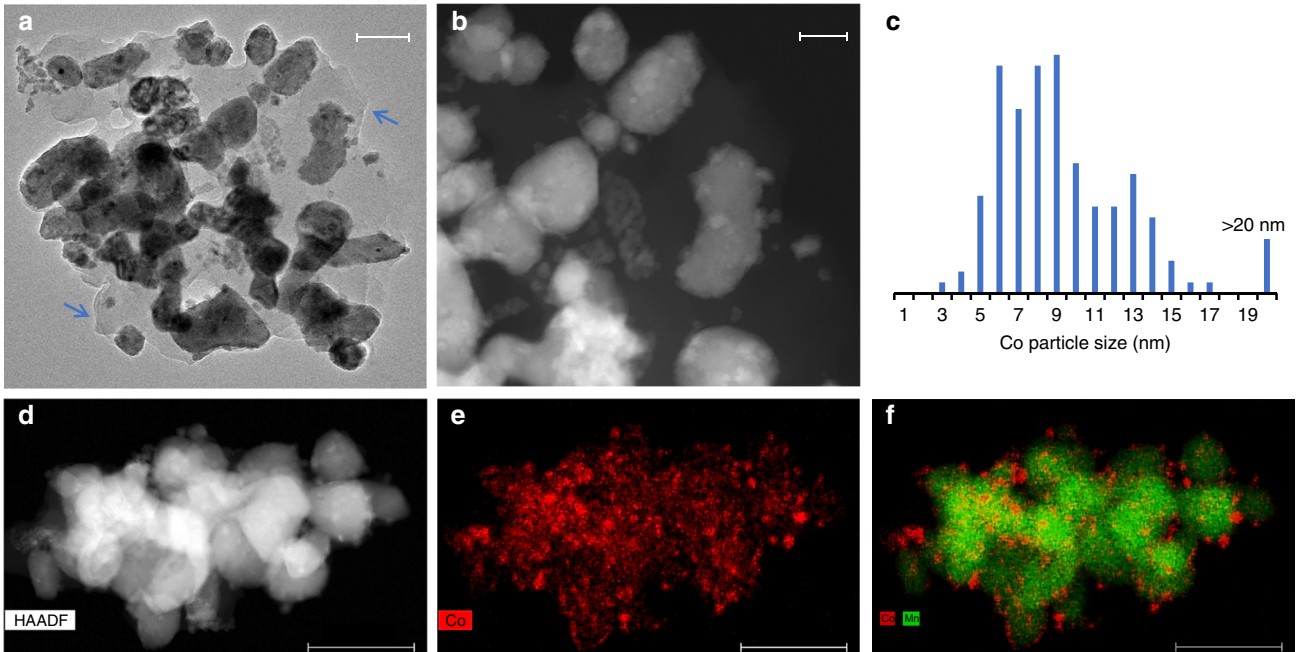

**Fig. 5** Electron microscopy images of spent $Co_1Mn_3–Na_2S$. Reaction conditions: 240–280 °C, 10 bar, and $H_2/CO = 2$. **a** bright-field TEM image with a scale bar corresponding to 100 nm, and blue arrows to point out the presence of wax, **b** dark-field TEM images with a scale bar corresponding to 50 nm, **c** particle size distribution of Co nanoparticles supported on MnO, and **d–f** STEM-EDX maps of Co and Mn, and the scale bars correspond to 200 nm

in agreement with the Co crystallite size of 9.2 nm with a standard deviation of 1.9 nm from XRD analysis. The elemental maps of Co and Mn in Fig. 5f confirmed that spent $Co_1Mn_3–Na_2S$ consisted of Co nanoparticles well dispersed on the MnO support.

**Theoretical calculations on $Na_2S$ vs. $Na_2O$.** To gain further understanding of the difference in $Na_2S$ and $Na_2O$, DFT calculations of both species on metallic Co (0001) surface were performed. Please note that these calculations are of a preliminary nature and further work is needed to arrive at full reaction pathway analysis which is outside the scope of this work. Pederson et al. recently performed DFT calculations on CoMnO systems for the production of light olefins and they found that selectivities can be attributed to an inhibited hydrogenation activity demonstrated by the increased barriers for $CH_3$ and $CH_4$ formation[22]. Strømsheim et al. recently showed that the restructuring of Co surface under CO exposure with K pre-adsorbed proceeded on the terraces rather than from the step edges[38]. Other notable theoretical studies on multiple elemental surfaces include ZnO/Cu, $Co_2C$/Co, Cu/Co[39–41]. While these studies are highly relevant, they are insufficient to explain current findings. As it is shown that the combination of $Na_2S$ is critical for product selectivity, the theoretical calculations were focused on $Na_2S$ and $Na_2O$ promotion. The function of the sodium promoter, as any alkali metal promoter, is to donate charge to the cobalt metal. For the manganese-containing catalysts studied here, this turns out to increase olefin formation. However, for good effect another counter-ion is needed, i.e., sulfur. As shown by DFT calculations (Fig. 6), the function of the sulfur promoter is to increase the charge donation from the Na promoter ions to the cobalt surface. When no specific counter-ions are added, sodium binds in the form of $Na_2O$ and a considerable part of the sodium charge donation is taken up by the oxygen atom. With sulfur it is suggested to form $Na_2S$ instead, resulting in a higher charge donation to the cobalt surface. The DFT calculations show that every $Na_2O$ moiety donates a total charge of −0.51 to the

cobalt surface (Na becomes +0.39 and O becomes −0.28), whereas $Na_2S$ donates a total charge of −0.62 (Na becomes +0.36 and S becomes −0.10). We tentatively interpret these results of higher charge donation to coincide with lower hydrogen coverages thus leading to lower methane selectivity in FTS similar to what we have reported for iron carbide[29].

**Structure-performance relations of metallic Co vs. $Co_2C$.** In Table 1, $CO_2$ selectivity was negligible for most catalysts except $Co_3Mn_1–Na_2O$ and $Co_3Mn_1–Na_2S$. From detailed XRD structural analysis of the spent catalysts, it was revealed that the active Co phase in $Co_1Mn_3$ and $Co_1Mn_3–Na_2S$ was metallic Co, but $Co_2C$ was present as an active phase in $Co_3Mn_1–Na_2O$. For classic Co-based FT catalysts (i.e. bulk Co and CoMn) with appropriate reduction/ activation procedure and reaction conditions, metallic Co is widely accepted to be the active phase[16,42]. As metallic Co catalysts are not active for WGS, it was expected that no $CO_2$ selectivity was observed for these catalysts. Upon the addition of $Na_2O$ or $Na_2S$, the ratio of Co/Mn apparently played a critical role in influencing the crystal structure of the Co phase during FT as $CO_2$ selectivities were significantly higher for $Co_3Mn_1$ than $Co_1Mn_3$. Li et al. recently concluded that Mn has a controlling effect on $Co_2C$ morphology and the formation of $Co_2C$ nanoprisms or nanospheres was dependent on the synthesis method[43]. In this study, the results of $Co_3Mn_1–Na_2O$ and $Co_3Mn_1–Na_2S$ were in agreement with Li et al. as Co content was higher than Mn in both cases. However, for catalysts with more Mn than Co, $Co_2C$ was not formed and Co remained in metallic phase. It is believed that when Co/Mn ≈ 0.3, $MnO_x$ served as a support for the metallic Co nanoparticles thereby ensuring a good dispersion and stabilization of these nanoparticles (Fig. 5). MnO is also known to act as an electronic and structural promoter and the promoting effects of MnO are strongly dependent on its location and amount. For instance, Morales et al. showed that CO preferentially bonded linearly to surface metal sites when MnO loading was increased[44]. It is noted, however, that in mentioned literature, the $MnO_x$ loading was much lower than the Co loading

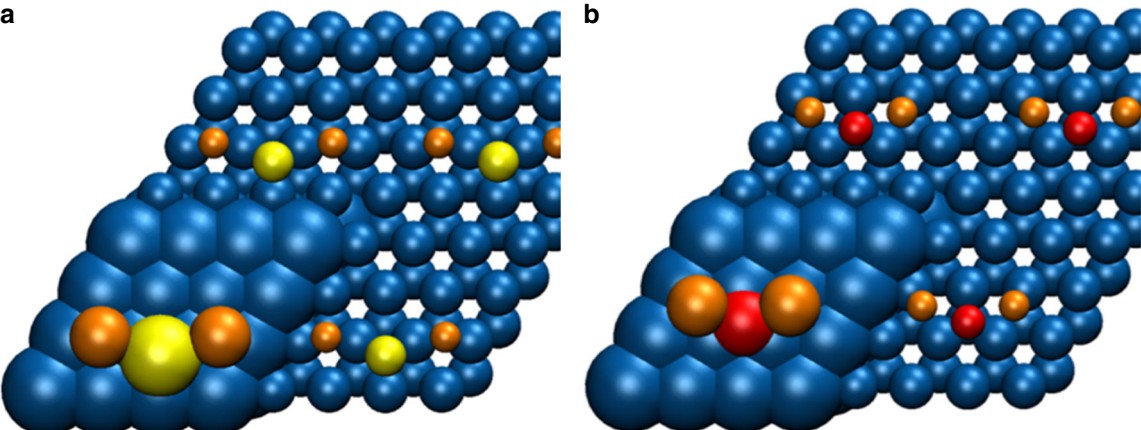

**Fig. 6** DFT-calculated binding geometries of $Na_2S$ and $Na_2O$ on the Co (0001) surface. **a** $Na_2O$ and **b** $Na_2S$ bind in a very similar fashion, although the O atom ends up above a subsurface cobalt atom and the S atom above an empty site. Atoms outside the calculation unit cell are depicted as smaller spheres; blue is Co, orange is Na, yellow is S and red is O

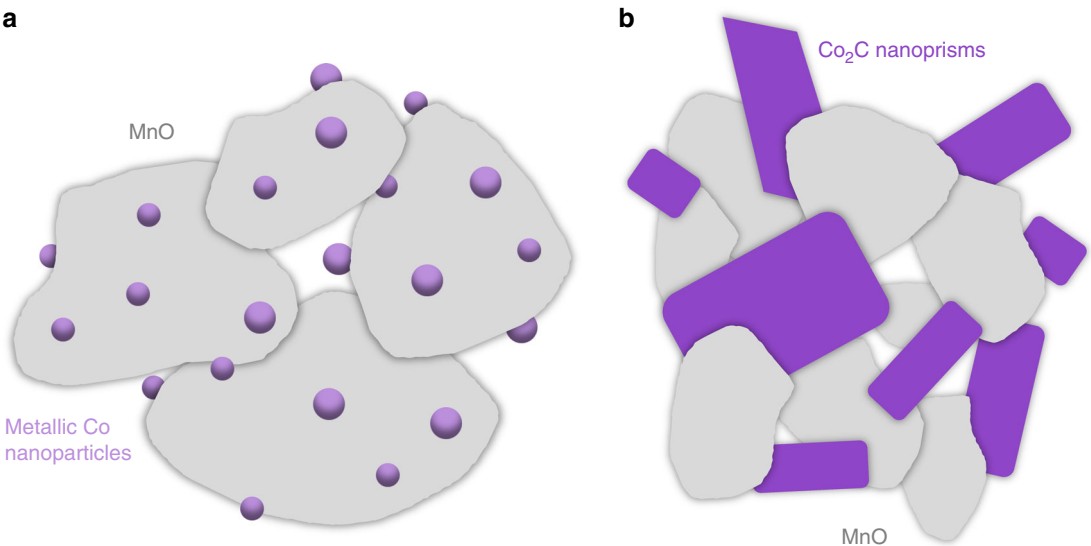

**Fig. 7** Schematic drawing of the structure of the active catalysts based on TEM and XRD analysis. **a** $Co_1Mn_3–Na_2S$ consists of metallic Co nanoparticles of ~10 nm dispersed on MnO support and **b** $Co_3Mn_1–Na_2O$ consists of $Co_2C$ nanoprisms of ~10–50 nm from Zhong et al[8]. Whereas the former catalyst restricts WGS and thus $CO_2$ formation, the latter leads to large amounts of $CO_2$

and the promoting effects of using MnO as a support are not yet clear. These findings are illustrated in Fig. 7.

While similar hydrocarbon product selectivities were reported earlier by Sun et al. for Na-promoted $Co_2Mn_1$ catalytic systems[8,23], it is noteworthy to point out that in their work $CO_2$ selectivity was almost 50% of CO converted due to high WGS activity. The addition of Na (most likely in $Na_2O$ state) served as a structural promoter and appeared to facilitate the formation of $Co_2C$ nanoprisms, which displayed high $C_2–C_4$ olefins and $CO_2$ selectivities. In our $Co_1Mn_3–Na_2S$ catalytic system, the active phase appeared to be metallic Co and $Na_2S$ seemed to be an electronic promoter for product selectivity. Sun et al. showed the effect of $Na_2O$ loading on $Co_2Mn_1$ catalytic systems, and here we presented the importance of the counter-ion for Na (Fig. 6) using theoretical DFT calculations.

Besides the loadings and counter-ions for Na, the activation procedure is an important parameter for catalytic performance and structure-performance relations. For instance, de Smit et al. demonstrated that different Fe-carbide phases may be synthesized during catalyst pretreatment by controlling carbon chemical potential[45]. Claeys et al. concluded that while cobalt carbide is relatively stable at typical reaction conditions, it would decompose rapidly into hcp Co with hydrogen at 150 °C[46]. Davis et al. also showed that reaction conditions played a significant role in formation of cobalt carbide or metallic cobalt[47]. To induce the formation of metallic Co, calcined catalysts in our study were reduced at 350 °C and 1 bar under diluted $H_2$ flow for 8 h, followed by introduction of syngas at a temperature of 180 °C and a pressure of 10 bar. This activation procedure to form metallic Co is different from that of Sun et al. to form $Co_2C$[8]. To show the effect of activation procedure on catalytic performance, the same catalysts were reduced at 300 °C and 1 bar under diluted $H_2$ flow for 5 h, followed by introduction of syngas at 250 °C and 10 bar. With this activation procedure, $CO_2$ selectivity increased to 6% (Supplementary Table 13) possibly related to cobalt carbide formation. Nonetheless, $Na_2S$ was still the most effective promoter (Supplementary Tables 11–13).

In summary, we have designed a catalytic system $Co_1Mn_3–Na_2S$, which showed negligible WGS activity and

suppression of methane formation in FTS. Structure analysis of the spent catalyst revealed 10 nm metallic Co nanoparticles as the active phase supported on MnO during reaction. Theoretical calculations revealed the importance of counter-ion S for Na, and $Na_2S$ was more efficient in tuning hydrocarbon product selectivity than $Na_2O$. Tentatively the addition of $Na_2S$ to $Co_1Mn_3$ was proposed to deactivate sites for secondary olefin hydrogenation and isomerization and for methanation, whereas the lower degree of alkalization as compared with $Na_2O$ is insufficient to promote the WGS reaction. For this complex catalytic system, further studies on the effects of various elements on structure-performance relations and advanced characterization are advocated.

The state-of-the-art processes and catalysts for direct production of lower olefins from synthesis gas are compared in Supplementary Table 14. While all catalysts showed favourable selectivity towards lower olefins, $Co_1Mn_3–Na_2S$ is the only catalyst, which combined lower olefin selectivity with negligible $CO_2$ production. This comparison suggests that $Co_1Mn_3–Na_2S$ is a promising catalyst which is capable of producing chemicals and fuels directly from $H_2$-rich syngas derived from natural gas. This gas-to-chemicals process would greatly reduce $CO_2$ emissions, thereby contributing prevention of climate change.

## Methods

**Synthesis of CoMn catalysts.** Two grams of $Co(NO_3)_2 \cdot 6H_2O$ (99 +%, Acros) and 5.7 g $Mn(NO_3)_2 \cdot 4H_2O$ (97.5 +%, Acros), were dissolved in 40 mL deionized water at room temperature in a 100 mL round-bottom flask. After 1 h of stirring at room temperature, the round-bottom flask was heated to 60 °C in a water bath. Twenty microliters of 1.0 M aqueous $(NH_4)_2CO_3$ (30 +% ($NH_3$), Acros) was added dropwise to the mixed nitrate solution using a mechanical pump set at 1 mL/min and pH was kept at ~8. The resulting pink powder was aged for 30 min at room temperature, followed by decanting and washing with deionized water thrice. The precipitate was then dried at 120 °C under static air for 2 h with stirring every 0.5 h and calcined at 400 °C (2 °C/min) under air flow for 2 h. The synthesized $Co_1Mn_3$ was then impregnated with either $Na_2CO_3$ anhydrous (99.5%, Fisher Scientific), $(NH_4)_2SO_4$ (≥99.0%, Sigma-Aldrich), $Na_2S_2O_3$ anhydrous (≥98.0%, Sigma-Aldrich) or $Na_2S$ nonahydrate (≥98.0%, Sigma-Aldrich) precursor's dissolved in deionized water, followed by calcination at 400 °C (2 °C/min) under air flow for 2 h. The $Co_3Mn_1$ catalysts were synthesized with the identical procedure but different Co and Mn precursors mass loadings.

**Catalyst characterization.** Elemental loading of Co, Mn, Na and S were determined with a Thermo Jarrell Ash model ICAP 61E trace analyzer inductively coupled plasma-atomic emission spectrometer (ICP-AES). Scanning electron microscopy (SEM) images were taken using a FEI XL30 FEG SEM instrument in backscattering electron mode at an acceleration voltage of 15 kV. SEM samples were prepared on carbon grids followed by Pt-coating to improve electron conductivity. STEM-HAADF images and EDX analysis were obtained with an FEI Talos F200X transmission electron microscope, operated at 200 kV and equipped with a high-brightness field emission gun (X-FEG) and a Super-X G2 EDX detector. More than 150 particles were measured to obtain a particle size distribution. XRD patterns were measured with a Bruker D8 Discover instrument in Debye-Scherrer transmission (capillary) geometry with a Mo ($K_{α1}$ 0.709 Å) source. A Göbel-mirror was used to focus a near-parallel X-ray beam on the 1000 μm (OD, wall thickness 10 μm) capillary. Energy dispersive LynxEye XE Position Sensitive Detector (PSD) was used, only accepting diffracted X-ray photons originating from Mo $K_α$ emission lines. Details on the instrument can be found in a recent publication[48]. Measurement parameters used were 2θ 5–48° with step size of 0.032° and exposure time of 18 s per step, for each measurement. Rietveld Quantitative Phase Analysis (Rietveld QPA) was performed on the measured diffractograms using Bruker TOPAS (v5) software. Details and discussion on the Rietveld refinement procedure are given in Supplementary Methods. Phase identification from diffractograms was done using ICDD PDF-4+2016 database and structures used in the Rietveld QPA were obtained from the same database and are listed in Supplementary Table 10.

**Catalytic tests at mild conditions.** Low pressure tests were carried out at 240 °C, 1 bar, $H_2/CO = 2$ v/v, <3% CO conversion. A fixed-bed reactor was loaded with 0.02 g (75–150 μm) catalyst and 0.20 g SiC (212–425 μm) for bed dilution. The catalysts were reduced prior to reaction at 350 °C (5 °C /min) under diluted $H_2$ flow (33 vol.% $H_2$, 67 vol.% He, 60 mL/min total flow) for 2 h. After reduction, temperature was decreased to 240 °C (2 °C /min) under 40 mL/min He flow. At 240 °C and 1 bar, the feed flow was switched to a mixture of $H_2$ and CO ($H_2/CO = 2$ v/v, 9

mL/min total flow). Hydrocarbons ($C_1–C_{16}$) from the product stream were analysed online with gas chromatography (Varian CP3800), and $CO_2$ was not measured. The line from the reactor to GC was heated to at least 150 °C to prevent hydrocarbon condensation. Activities and product selectivities were calculated on a carbon atom basis. Activity is reported as moles of CO converted per gram Co per second, and moles of CO converted is based on moles of C in the hydrocarbon product stream. Product selectivity was calculated as equivalent of carbon atoms in a product with respect to the total carbon atoms present in the hydrocarbons produced (% C).

**Catalytic tests at industrially relevant conditions.** Medium pressure tests were performed using a high throughput 16 parallel fixed-bed reactors set-up (Flowrence, Avantium). Each reactor was loaded with 50 mg catalyst (75–150 μm) and 100 μL SiC (212–425 μm) as diluent. The catalysts were first dried at 100 °C (5 °C /min) under He flow for 2 h and subsequently reduced at 350 °C (1 °C/min) under dilute $H_2$ flow (25 vol.% $H_2$, 75 vol.% He) for 8 h. After reduction, temperature was decreased to 180 °C (1 °C /min) and pressure was increased to 10 bar under $H_2$ flow. At 180 °C and 10 bar, the feed flow was switched to syngas mixture ($H_2/CO/He = 60/30/10$, 6.6 mL/min total flow per reactor) and subsequently the temperature was raised to 240 °C (1 °C /min). The product stream was analysed using online gas chromatography (Agilent 7890A) with Ar as carrier gas. Hydrocarbons ($C_1–C_9$) were separated on an Agilent J&W PoraBOND Q column, detected using an FID detector and quantified against the TCD signal of the internal standard He. The permanent gases (CO, $H_2$, He, $CO_2$ and $CH_4$) were separated on a ShinCarbon ST (#19043) column and quantified against He as an internal standard using a TCD detector. $CO_2$ was also measured and the detection limit of $CO_2$ was determined to 0.5% yield, which was 3% $CO_2$ selectivity and 18% CO conversion (Supplementary Figure 5 and Supplementary Table 8). Catalytic activity and product selectivities were measured at 240–280 °C, 10 bar, $H_2/CO = 2$, 10–70% CO conversion. To show the effect of activation procedure on catalytic performance, the same catalysts were reduced at 300 °C and 1 bar under diluted $H_2$ flow for 5 h, followed by introduction of syngas at 250 °C and 10 bar. Definitions of the selectivity and activity, expressed as CO conversion and cobalt-time-yield (CTY) are included as Supplementary Methods.

**DFT calculations.** DFT modelling was performed with the ADF-BAND package (version 2016.102)[49,50], using the rPBE functional[51] and Grimme D3 corrections[52]. A TZP basis set with small frozen cores, a "good" k-space, and otherwise "normal" settings were used. For efficiency, the SCF was converged to only $5 \times 10^{-4}$ Hartree. Gradients were converged to 0.001 Hartree/Å. The bulk cobalt unit cell vectors were reoptimized, giving $a = 2.43$ Å (experimental 2.51 Å) and $c = 3.91$ Å (experimental 4.07 Å). The (0001) surface was modelled with 6 atomic layers, giving a slab of 12 Å thick, of which the bottom two layers were frozen and calculated at minimal settings (SZ basis set with large frozen core, orbital confinement to 4 bohr, and "basic" settings for the Becke grid and zlm-fit parameters). The surface unit cell consisted of $4 \times 4$ atoms. Since ADF-BAND uses true 2D periodicity, no vacuum spacing nor dipole corrections were needed. Atomic charges were calculated with Hirshfeld's method[53].

## Data availability

The datasets generated during and/or analysed during the current study are available from the corresponding author on reasonable request.

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

## Acknowledgements

This research received funding from the Netherlands Organisation for Scientific Research (NWO) in the framework of the TASC Technology Area "Syngas, a Switch to Flexible New Feedstock for the Chemical Industry (TA-Syngas)". Dow and Johnson Matthey (JM) are also acknowledged for the funding received. K.P.d.J. acknowledges the European Research Council, EU FP7 ERC Advanced Grant no. 338846. J.J. Mulder (ICP-AES), Y. Wei ($N_2$-physisorption) M. Versluijs-Helder (SEM-EDX), and H. Meeldijk (TEM-EDX) are acknowledged for the respective measurements. Dr. M. Ruitenbeek (Dow), Dr. M. Watson (JM) and Dr. L. van der Water (JM) are thanked for fruitful discussions.

## Author contributions

K.P.d.J and K.P.d.J conceived, coordinated the research and designed the experiments. J.X. synthesized, characterized and tested catalysts. T.W.v.D contributed to the design of experiments and evaluation of catalyst performance. B.M.W. and P.P. conceptualized and performed XRD characterization, including Rietveld analysis. M.J.L. performed DFT calculations. All authors contributed to analysis and discussion on the data. The manuscript was primarily written by J.X. and K.P.d.J with input from all authors.

## Additional information

**Competing interests:** The authors declare no competing interests.

