## [Peer Review File · Nature Communications]

Reviewers' Comments:

Reviewer #1:

Remarks to the Author:

The manuscript accomplished by Xie et al. reported a Co₁Mn₃-Na₂S catalyst consisting of Co/Mn/Na/S, which showed high C₂-C₄ olefins selectivity with relative low WGS activity, and the products distribution deviates from ideal ASF distribution especially for the C₁ and C₂. The authors completed a successful work to improve the selectivity to lower olefins using metallic Co-based catalyst while the formation of methane was suppressed. The authors have revised the manuscript based on the reviewer's comments. However, the authors still need to consider the following comments:

(1) The topic is the direct production of C₂-4 olefins with high selectivity with few CO₂ in this manuscript and fuels or other chemicals are not mentioned. I suggest the authors modify the former title to better reflect the research topic of this study.

(2) The loading amount of Na₂S was very low for the studied catalyst. In order to further reveal the promoted effect of Na₂S, a series catalyst with different loading amount of Na₂S should be investigated. What is the best loading amount?

(3) The authors show an interesting FTO catalyst but do not provide enough evidence to explain the underlying mechanism. Why the addition of S and Na can hinder the secondary olefin hydrogenation or methanation formation? The only evidence from DFT calculations offers little insight and cannot convince me. The authors also supposed that higher charge donation to coincide with lower hydrogen coverages lead to lower methane selectivity (in similar to what they have reported for iron carbide). However, it is just a speculation. The authors should provide specific experimental data.

(3) At mild conditions of 240 °C, 1 bar, H₂/CO=2, 1% CO conversion, the authors said the olefins selectivity was 54% with a C₂-C₄ olefin/paraffin ratio of 17. However, at such a low CO conversion, it is difficult to ensure the accuracy of the reaction data, including the CO conversion and products selectivity. I noticed that a Varian CP3800 GC was used to analyze the hydrocarbons (C₁-C₁₆). Was the offgas kept warm from the reactor to the GC? Why did the authors neglect the hydrocarbons with carbon number larger than 16? I noticed there existed diffraction peaks for wax phase on the spent catalysts (in XRD analysis).

(4) I suggest the authors provide the detailed reaction conditions in the legend for Figure 1a (pressure), 1b (temperature) and 1c (temperature).

(5). In Supplementary Figure 6a, it should be "Co₁Mn₃-Na₂S" instead of "CO₁Mn₃-Na₂S"?

Reviewer #2:

Remarks to the Author:

I have read the authors responds to the reviewers requests and the changes made and propose publication of the revised manuscript.

Perhaps, some of the newly added information could have been presented in the main article (instead among further information in the Internet) as thinking them actually relevant.

Reviewer #3:

Remarks to the Author:

Thank the author for their time to reply to my comments. Despite this, I remain unsatisfied with the theoretical DFT calculations in this work and cannot recommend it for publication in its current state. Another reason that I hesitate to suggest its publication is that the FT process is not of significant interest to a general scientific audience unless a very major breakthrough is made.

Detailed rebuttal:

*Text in italics: Comments of the reviewers*

**Blue text:** Rebuttal text

**Red text:** Main text of the manuscript, **bold indicates text changes**

**Green text:** Supporting information, **bold indicates text changes**

Reviewers' Comments:

Reviewer #1 (Remarks to the Author):

*The manuscript accomplished by Xie et al. reported a Co1Mn3-Na2S catalyst consisting of Co/Mn/Na/S,*
*which showed high C2-C4 olefins selectivity with relative low WGS activity, and the products distribution*
*deviates from ideal ASF distribution especially for the C1 and C2. The authors completed a successful work*
*to improve the selectivity to lower olefins using metallic Co-based catalyst, while the formation of methane*
*was suppressed. The authors have revised the manuscript based on the reviewer's comments. However, the*
*authors still need to consider the following comments:*

**Author Reply:** We thank the reviewer for his/her critical, but positive review of our work and we aim to
improve our manuscript using this feedback.

*(1) The topic is the direct production of C2-4 olefins with high selectivity with few CO2 in this manuscript*
*and fuels or other chemicals are not mentioned. I suggest the authors modify the former title to better reflect*
*the research topic of this study.*

**Author Reply:** We agree with the reviewer and have changed the title of the manuscript to “Promoted Cobalt
Metal Catalysts Suitable for the Production of Lower Olefins from Natural Gas”

*(2) The loading amount of Na2S was very low for the studied catalyst. In order to further reveal the*
*promoted effect of Na2S, a series catalyst with different loading amount of Na2S should be investigated.*
*What is the best loading amount?*

**Author Reply:** We agree that a series of catalysts with different loading amount of Na₂S is useful for further
optimization of our new catalysts but it was not within the scope of current study. We referred to previous
literature, in particular the work of Visconti et al. (ref. 32) on effect of S on Co/Al₂O₃ FT catalysts, for a
starting point of S loading and modified our catalysts accordingly. Nonetheless, we have varied the
precursor of Na/S and loading of Na and found that the favourable effects on selectivity remain. The ICP
analysis of Co₁Mn₃-Na₂S₂O₃ is now included in Supplementary Table 1 and its catalytic performance in
Figure 2b, Table 1, Supplementary Tables 6, 7, 9, 11, 12, 13 and Supplementary Figure 7.

**Co₁Mn₃ catalysts with an atomic ratio Co/Mn=1/3 were synthesized via co-precipitation, and the calcined**
**catalysts were impregnated with Na₂CO₃, (NH₄)₂SO₄, Na₂S₂O₃ or Na₂S precursors followed by another**
**calcination step. These catalysts were named Co₁Mn₃-Na₂O, Co₁Mn₃-SO₄²⁻, Co₁Mn₃-Na₂S₂O₃ and**

**Co₁Mn₃-Na₂S, respectively.** As a comparison, Co₃Mn₁ catalysts were also synthesized and named in a
 similar fashion.

The synthesized Co₁Mn₃ was then impregnated with either **Na₂CO₃ anhydrous (99.5%, Fisher Scientific),**
 **(NH₄)₂SO₄ (≥99.0%, Sigma Aldrich), Na₂S₂O₃ anhydrous (≥98.0%, Sigma Aldrich) or Na₂S**
 **nonahydrate (≥98.0%, Sigma Aldrich) precursors dissolved in deionized water,** followed by
 calcination at 400 °C (2 °C/min) under air flow for 2 h.

Supplementary Table 1

List of catalysts with their elemental loadings and atomic ratios measured with ICP-AES.

	% Weight				Atomic Ratio			
	Co	Mn	Na	S	Mn/Co	S/Co	Na/Co	Na/S
Co	77.75	0.00	0.01	0.00				
Mn	0.00	68.77	0.02	0.12				
Co ₃ Mn ₁	58.38	16.23	0.03	0.03	0.30	0.00	0.00	1.06
Co ₃ Mn ₁ -Na ₂ O	57.91	16.13	0.25	0.03	0.30	0.00	0.01	10.40
Co ₃ Mn ₁ -Na ₂ S	57.27	16.04	0.24	0.21	0.30	0.01	0.01	1.61
Co ₁ Mn ₃	15.03	52.38	0.02	0.09	3.74	0.01	0.00	0.23
Co ₁ Mn ₃ -Na ₂ O	15.55	54.21	0.07	0.09	3.74	0.01	0.01	1.04
Co ₁ Mn ₃ -Na ₂ S	15.81	54.91	0.08	0.14	3.73	0.02	0.01	0.78
Co₁Mn₃-Na₂S₂O₃	15.54	52.59	0.04	0.13	3.63	0.02	0.01	0.45

Figure 2: a) Catalytic performance (%) over time of Co₁Mn₃-Na₂S and (b) Cobalt-time-yield (CTY) over
 time of various Co-based catalysts at 240 °C, 3 bar, and H₂/CO = 2.

As shown in Figure 2b, the addition of Mn increased activity for Co-based catalysts, which is in agreement
 with literature.^{33,34} Catalysts with Co/Mn ≈ 0.3 showed highest activity per gram Co (CTY), and the addition
 of Na₂O, Na₂S₂O₃ or Na₂S decreased activity. Nonetheless, the activity of Co₁Mn₃-Na₂S was still higher
 than the remaining Co-based catalysts. In terms of stability, Co₁Mn₃, Co₁Mn₃-Na₂O, Co₁Mn₃-Na₂S₂O₃ and
 Co₃Mn₁ showed deactivation but all other catalysts remained stable over 70 h.

Table 1: Catalytic performance at 240 °C, 10 bar, H₂/CO = 2, 18 – 30 % CO conversion. The detection limit
 of CO₂ selectivity is 0.5% yield, equivalent to 3 % CO₂ selectivity at 18 % CO conversion.

	CO conv. (%)	CTY (10 ⁻⁴ mol _{CO} /g _{Co} /s)	Product Selectivity (%C)					O/P C ₂ -C ₄	α
			C ₁	C ₂ -C ₄ olefins	C ₂ -C ₄ paraffins	C ₅₊	CO ₂		
Co	32	0.13	12	6	7	75	< 2	0.8	0.69
Co ₃ Mn ₁	31	0.14	11	17	10	62	< 2	1.8	0.63
Co ₃ Mn ₁ -Na ₂ O	20	0.09	9	14	7	42	28	1.9	0.53
Co ₃ Mn ₁ -Na ₂ S	25	0.12	5	20	7	56	13	2.9	0.56
Co ₁ Mn ₃	31	0.56	15	12	13	61	< 2	0.9	0.67
Co ₁ Mn ₃ -Na ₂ O	27	0.46	14	11	12	63	< 2	0.9	0.65
Co ₁ Mn ₃ -Na ₂ S	18	0.40	4	30	7	59	< 3	4.2	0.53
Co₁Mn₃-Na₂S₂O₃	22	0.42	7	25	12	56	<3	2.1	0.50

 At more industrially relevant conditions of 240 °C, 10 bar, H₂/CO = 2, 18 – 30 % CO conversion,
 the catalytic performance of Co₁Mn₃-Na₂S was compared to other Co-based catalysts (Table 1). Co₁Mn₃-
 Na₂S displayed the highest selectivity towards lower olefins and lowest selectivities towards undesired
 methane and lower paraffins (C₂-C₄ olefin/paraffin ratio = 4.2). Remarkably, CO₂ selectivity was below 3
 71 %C (below detection limit, see Supplementary Figure 6 for chromatograms), suggesting the absence of
 72 WGS activity and making it an attractive catalyst for H₂-rich syngas. CO₂ selectivity was consistently below
 73 detection limit for all catalysts except where less Mn is present, i.e. Co₃Mn₁-Na₂O and Co₃Mn₁-Na₂S. Even
 so, the suppression of WGS activity by Na₂S instead of Na₂O addition was evident by the CO₂ selectivity
 of Co₃Mn₁-Na₂S compared to Co₃Mn₁-Na₂O, i.e. 13 versus 28 %C, respectively. **The precursor of Na/S**
 **and loading of Na were varied (Na₂S and Na₂S₂O₃) and the favorable effects on selectivity remain**
 **(Supplementary Table 1 and Table 1). Further optimization of precursor and loadings of the**
 **promoters is however outside the scope of this study.**

Supplementary Table 6

Catalytic performance at 240 °C, 3 bar, H₂/CO = 2.

	CO conv. (%)	CTY (10 ⁻⁴ mol _{CO} /g _{Co} /s)	Product Selectivity (%C)					O/P C ₂ -C ₄	α
			C ₁	C ₂ -C ₄ olefins	C ₂ -C ₄ paraffins	C ₅₊	CO ₂		
Co	10	0.04	29	11	19	41	<5	0.6	0.51
Co ₃ Mn ₁	29	0.14	16	15	8	61	<2	1.9	0.68
Co ₃ Mn ₁ -Na ₂ O	10	0.05	6	15	4	42	33	3.9	0.50
Co ₃ Mn ₁ -Na ₂ S	15	0.08	6	26	6	47	14	4.1	0.52
Co ₁ Mn ₃	37	0.68	11	14	7	68	<2	1.9	0.71
Co ₁ Mn ₃ -Na ₂ O	21	0.36	9	18	6	67	<3	2.8	0.66
Co ₁ Mn ₃ -Na ₂ S	10	0.20	7	42	7	43	<5	5.6	0.48
Co₁Mn₃-Na₂S₂O₃	15	0.28	9	28	9	54	<3	3.1	0.56

Supplementary Table 7

Catalytic performance at 240 °C, 5 bar, H₂/CO = 2.

	CO conv. (%)	CTY (10 ⁻⁴ mol _{CO} /g _{Co} /s)	Product Selectivity (%C)					O/P C ₂ -C ₄	α
			C ₁	C ₂ -C ₄ olefins	C ₂ -C ₄ paraffins	C ₅ +	CO ₂		
Co	15	0.06	22	10	16	52	<3	0.6	0.65
Co ₃ Mn ₁	31	0.14	12	15	7	65	<2	2.1	0.68
Co ₃ Mn ₁ -Na ₂ O	13	0.06	8	15	5	42	30	2.8	0.5
Co ₃ Mn ₁ -Na ₂ S	19	0.09	5	22	6	50	16	3.7	0.54
Co ₁ Mn ₃	33	0.61	11	14	9	66	<2	1.5	0.69
Co ₁ Mn ₃ -Na ₂ O	22	0.38	16	13	10	62	<3	1.3	0.66
Co ₁ Mn ₃ -Na ₂ S	13	0.29	5	33	7	56	<4	5.1	0.51
Co₁Mn₃-Na₂S₂O₃	18	0.35	7	29	9	55	<3	3.2	0.52

Supplementary Figure 7

Fraction of C₄ products produced by different catalysts at 240 °C, 10 bar, H₂/CO = 2, 18 – 30 % CO
 conversion.

Supplementary Table 9

Fraction of C₄ products produced by different catalysts at varied reaction conditions, extension of Table 1,
 Supplementary Table 4, 5, 6 and 7.

	Pressure (bar)	Temp. (°C)	Fraction of C ₄ products		
			n-butane	1-butene	2-butene
Co	3	240	0.63	0.29	0.09
	5	240	0.64	0.29	0.07
	10	240	0.47	0.48	0.05
Co ₃ Mn ₁	3	240	0.26	0.66	0.08
	5	240	0.26	0.69	0.04
	10	240	0.30	0.66	0.04
Co ₃ Mn ₁ -Na ₂ O	3	240	0.13	0.81	0.06
	5	240	0.18	0.75	0.07
	10	240	0.31	0.63	0.06
Co ₃ Mn ₁ -Na ₂ S	3	240	0.14	0.76	0.11
	5	240	0.16	0.75	0.09
	10	240	0.25	0.70	0.06
Co ₁ Mn ₃	3	240	0.27	0.60	0.13
	5	240	0.33	0.58	0.10
	10	240	0.47	0.46	0.07
	10	260	0.82	0.10	0.08
	10	280	0.73	0.13	0.14
Co ₁ Mn ₃ -Na ₂ O	3	240	0.19	0.72	0.09
	5	240	0.36	0.53	0.11
	10	240	0.48	0.44	0.08
Co ₁ Mn ₃ -Na ₂ S	3	240	0.11	0.81	0.07
	5	240	0.16	0.79	0.05
	10	240	0.19	0.78	0.04
	10	260	0.32	0.61	0.07
	10	280	0.23	0.55	0.22
Co₁Mn₃-Na₂S₂O₃	3	240	0.16	0.70	0.14
	5	240	0.16	0.73	0.10
	10	240	0.26	0.65	0.09

Supplementary Table 11
 Catalytic performance at 240 °C, 3 bar, H₂/CO = 2 with ‘carbide’ activation procedure.

	CO conv. (%)	CTY (10 ⁻⁴ mol _{CO} /g _{Co} /s)	Product Selectivity (%C)					O/P C ₂ -C ₄
			C ₁	C ₂ -C ₄ olefins	C ₂ -C ₄ paraffins	C ₅₊	CO ₂	
Co	28	0.09	20	8	17	55	0	0.4
Co ₃ Mn ₁	47	0.22	13	8	14	62	3	0.6
Co ₃ Mn ₁ -Na ₂ O	25	0.12	11	17	9	40	23	1.8
Co ₃ Mn ₁ -Na ₂ S	24	0.11	12	20	14	39	14	1.4
Co ₁ Mn ₃	33	0.59	8	14	6	72	0	2.5
Co ₁ Mn ₃ -Na ₂ O	16	0.28	10	20	8	52	10	2.5
Co ₁ Mn ₃ -Na ₂ S	17	0.28	7	30	7	51	5	4.3
Co₁Mn₃-Na₂S₂O₃	21	0.37	10	18	6	63	3	2.8

Supplementary Table 12
 Catalytic performance at 240 °C, 5 bar, H₂/CO = 2 with ‘carbide’ activation procedure.

	CO conv. (%)	CTY (10 ⁻⁴ mol _{CO} /g _{Co} /s)	Product Selectivity (%C)					O/P C ₂ -C ₄
			C ₁	C ₂ -C ₄ olefins	C ₂ -C ₄ paraffins	C ₅₊	CO ₂	
Co	27	0.09	21	8	17	54	0	0.5
Co ₃ Mn ₁	37	0.18	14	10	16	58	3	0.6
Co ₃ Mn ₁ -Na ₂ O	34	0.16	10	16	9	44	22	1.8
Co ₃ Mn ₁ -Na ₂ S	30	0.14	11	19	14	42	14	1.4
Co ₁ Mn ₃	31	0.56	16	11	10	61	2	1.1
Co ₁ Mn ₃ -Na ₂ O	18	0.30	14	17	10	49	10	1.6
Co ₁ Mn ₃ -Na ₂ S	21	0.35	6	27	8	54	5	3.6
Co₁Mn₃-Na₂S₂O₃	20	0.35	8	22	8	58	4	2.7

Supplementary Table 13
 Catalytic performance at 240 °C, 10 bar, H₂/CO = 2 with ‘carbide’ activation procedure.

	CO conv. (%)	CTY (10 ⁻⁴ mol _{CO} /g _{Co} /s)	Product Selectivity (%C)					O/P C ₂ -C ₄
			C ₁	C ₂ -C ₄ olefins	C ₂ -C ₄ paraffins	C ₅₊	CO ₂	
Co	45	0.16	15	6	10	70	0	0.6
Co ₃ Mn ₁	41	0.19	16	11	16	52	4	0.7
Co ₃ Mn ₁ -Na ₂ O	51	0.24	10	13	9	46	22	1.4
Co ₃ Mn ₁ -Na ₂ S	41	0.19	9	16	12	48	15	1.3
Co ₁ Mn ₃	29	0.51	16	10	13	62	0	0.8
Co ₁ Mn ₃ -Na ₂ O	22	0.38	12	13	12	54	9	1.1
Co ₁ Mn ₃ -Na ₂ S	30	0.51	5	23	8	58	6	3.0
Co₁Mn₃-Na₂S₂O₃	30	0.52	7	19	11	59	4	1.8

(3) The authors show an interesting FTO catalyst but do not provide enough evidence to explain the
underlying mechanism. Why the addition of S and Na can hinder the secondary olefin hydrogenation or
methanation formation? The only evidence from DFT calculations offers little insight and cannot convince
me. The authors also supposed that higher charge donation to coincide with lower hydrogen coverages
lead to lower methane selectivity (in similar to what they have reported for iron carbide). However, it is
just a speculation. The authors should provide specific experimental data.

Author Reply: Our catalyst system consists of 4 different elements and to the best of our knowledge, there
is no (limited) literature regarding theoretical calculations on such complicated systems. The latest state of
the art theoretical research consists of model systems of 3 elements, such as Kattel et al. (Kattel et al.,
*Science* 355, 1296 – 1299, 2017) who studied active sites for CO₂ hydrogenation on Zn/Cu vs ZnO/Cu, and
Pederson et al. (Pederson et al., *Journal of Catalysis* 361, 23 – 32, 2018) who performed DFT calculations
on CoMnO systems for Fischer-Tropsch production of light olefins. Other examples include Voss et al.
(Voss et al., *Topics in Catalysis* 61, 1016 – 1023, 2018) who calculated lowest DFT surface energies of
Cu/Co surface as a function of CO coverage and surface Co concentration. In the same publication, they
highlighted that less information is presently available about the virgin metallic phase for the CoMn-based
catalysts thus they did not carry out further theoretical calculations on such systems. The work of Zhong et
al. (ref. 8) reported on Co/Mn/Na catalysts and DFT calculations were performed on Co₂C and Co, which
were simplified models on their catalysts.

Moreover, there is limited literature on theoretical calculations of alkali oxide/sulfide and metallic cobalt
and we think that our calculations provided new insights in this area. For instance, Strømsheim et al.
recently investigated the effect of K adsorption on CO-induced restructuring of CO surface (Strømsheim et
al., *Catalysis Today* 299, 37 – 46, 2018) and such a thorough study is uncommon in literature. Any further
DFT calculations at this stage would require too many assumptions. This discussion is now added to the
manuscript as follows:

To gain further understanding of the difference in Na₂S and Na₂O, DFT calculations of both species on
metallic Co (0001) surface were performed. Please note that these calculations are of a preliminary nature
and further work is needed to arrive at full reaction pathway analysis which is outside the scope of this
work. **Pederson et al. recently performed DFT calculations on CoMnO systems for the production of
light olefins and they found that selectivities can be attributed to an inhibited hydrogenation activity
demonstrated by the increased barriers for CH₃ and CH₄ formation.²¹ Strømsheim et al. recently
showed that the restructuring of Co surface under CO exposure with K pre-adsorbed proceeded on
the terraces rather than from the step edges.³⁷ Other notable theoretical studies on multiple elemental
surfaces include ZnO/Cu, Co₂C/Co, Cu/Co.^{38–40} While these studies are highly relevant, it is
insufficient to explain current findings. As it is shown that the combination of Na₂S is critical for
product selectivity, the theoretical calculations were focused on Na₂S and Na₂O promotion.**

(3) At mild conditions of 240 °C, 1 bar, H₂/CO=2, 1% CO conversion, the authors said the olefins
selectivity was 54% C with a C₂-C₄ olefin/paraffin ratio of 17. However, at such a low CO conversion, it is
difficult to ensure the accuracy of the reaction data, including the CO conversion and products selectivity.
I noticed that a Varian CP3800 GC was used to analyze the hydrocarbons (C₁-C₁₆). Was the offgas kept
warm from the reactor to the GC? Why did the authors neglect the hydrocarbons with carbon number

*larger than 16? I noticed there existed diffraction peaks for wax phase on the spent catalysts (in XRD*
*analysis).*

*Author Reply:* We thank the reviewer for the question and the opportunity to clarify the experiments. Firstly,
we agree that it may be challenging to obtain absolute accuracy at very low CO conversions, especially if
CO conversion is to be calculated as $(\text{CO}_{\text{in}} - \text{CO}_{\text{out}})/\text{CO}_{\text{in}}$ as there will be numerous variables such as feed
flow measured with MFC, concentrations measured with GC etc. However, in our calculation definition for
such low CO conversions, CO conversion is determined by C concentrations of all products measured by
FID detector and the FID detector is highly accurate. Secondly, the offgas was kept warm from the reactor
to the GC, as the line from reactor to GC was heated to at least 150 °C to prevent hydrocarbon condensation.
Thirdly, while the FID detector was calibrated for C1-C16 hydrocarbons, no hydrocarbon peaks were
observed after C13 and there were very small amounts of C10 – C13. Thus, we do not think that
hydrocarbons with carbon number larger than 16 would be produced at mild conditions (240 °C and 1 bar).
Lastly, the XRD analysis of spent catalysts (Figure 3) refers to spent catalysts after 240 240 - 280 °C, 10
165 bar and $\text{H}_2/\text{CO} = 2$, whereby wax was observed in both XRD and TEM analysis. This discussion is now
added to the manuscript as follows.

**Figure 3. a) Background corrected XRD patterns and b) Rietveld QPA based crystalline phase compositions**
**of spent Co_1Mn_3 , $\text{Co}_1\text{Mn}_3\text{-Na}_2\text{S}$, $\text{Co}_3\text{Mn}_1\text{-Na}_2\text{O}$ after industrially relevant conditions of 240 - 280 °C, 10**
**bar, and $\text{H}_2/\text{CO} = 2$.**

**Hydrocarbons ($\text{C}_1\text{-C}_{16}$) from the product stream were analysed online with gas chromatography (Varian**
**CP3800), and CO_2 was not measured. The line from the reactor to GC was heated to at least 150 °C to**
**prevent hydrocarbon condensation.**

*(4) I suggest the authors provide the detailed reaction conditions in the legend for Figure 1a (pressure), 1b*
*(temperature) and 1c (temperature).*

*Author Reply:* We thank the reviewer for the suggestion and the legend for Figure 1 is modified according
to the suggestion.

**Figure 1: a) Product Selectivity (%C) of $\text{Co}_1\text{Mn}_3\text{-Na}_2\text{S}$ at different temperatures, 240 – 280 °C, 10 bar,**
**$\text{H}_2/\text{CO} = 2$, and b) Product Selectivity (%C) of $\text{CoMn}_3\text{-Na}_2\text{S}$ at different pressures, 3 – 10 bar, 240 °C,**
**$\text{H}_2/\text{CO} = 2$**

*(5). In Supplementary Figure 6a, it should be “ Co1Mn3-Na2S ” instead of “ CO1Mn3-Na2S ”?*

*Author Reply:* We thank the reviewer for identifying the typo and the wording is corrected accordingly.

**Reviewer #2 (Remarks to the Author):**

*I have read the authors responds to the reviewers requests and the changes made and propose publication*
*of the revised manuscript.*

*Perhaps, some of the newly added information could have been presented in the main article (instead*
*among further information in the Internet) as thinking them actually relevant.*

Author Reply: We thank the reviewer for his/her positive recommendation and comments. Accordingly,
we have moved Supplementary Figure 6 to Figure 3 in the main article. As there is an overlap of information
in Figure 1c and d with the new Figure 3, Figure 1c and d are shifted to Supplementary Information
(Supplementary Figure 5).

Reviewer #3 (Remarks to the Author):

*Thank the author for their time to reply to my comments. Despite this, I remain unsatisfied with the*
*theoretical DFT calculations in this work and cannot recommend it for publication in its current state.*
*Another reason that I hesitate to suggest its publication is that the FT process is not of significant interest*
*to a general scientific audience unless a very major breakthrough is made.*

Author Reply: We thank the referee for his/her response. However, we wish to re-emphasize that our DFT
calculations are of a preliminary nature. On the other hand, for the first time we report that Na+S on Co-
based FT catalyst leads to a breakthrough in providing high olefin selectivities with concomitant
suppression of the water gas shift reaction. The importance of decreasing CO₂ production during the FT
step was recently highlighted by Wang et al in their development of phase pure stable and low-CO₂ selective
ε-iron carbide FT catalysts for the coal-to-liquids process (Wang et al., *Sci. Adv.* 4, eaau2947, 2018). We
consider our research to be an important finding in this field of research, therefore believe that a multi-
disciplinary journal, such as *Nature Communications*, would be a good medium for the publication of this
work. This discussion is now included in the manuscript.

Deviation of the ASF distribution to suppress methane formation is critical to attain high fractions of lower
olefins, and this is possible with promoted Fe-carbide-based⁵⁻⁷ and promoted Co-carbide-based catalysts.^{8,9}
However, **most** carbide-based catalysts are also active for the Water-Gas-Shift (WGS) reaction,¹⁰ thereby
producing CO₂ and rendering them inefficient for methane-derived H₂-rich syngas. Similarly, the
bifunctional oxide-zeolite catalysts, which converts syngas directly to lower olefins, showed high activity
for WGS and are only suitable for coal-derived CO-rich syngas.^{11,12} **The importance of decreasing CO₂**
**production during the FT step was recently highlighted by Wang et al. in their development of phase**
**pure stable and low-CO₂ selective ε-iron carbide FT catalysts for the coal-to-liquids process.**¹³

Reviewers' Comments:

Reviewer #1:

Remarks to the Author:

The authors have addressed all the comments made by the referees and the manuscript has been improved. In my opinion the paper is now ready for publication.

REVIEWERS' COMMENTS:

Reviewer #1 (Remarks to the Author):

The authors have addressed all the comments made by the referees and the manuscript has been improved. In my opinion the paper is now ready for publication.

Author Response: We thank the reviewer for his/her positive recommendation.